# OpenReview forum: "Latent Sentinel: Real-Time Jailbreak Detection with Layer-wise Probes"
_ICLR.cc/2026/Conference — ICLR 2026 Conference Withdrawn Submission_

### Official Review · Reviewer_W3Q2 · 2025-10-27

**Soundness:** 3
**Presentation:** 3
**Contribution:** 2
**Rating:** 2
**Confidence:** 4

**Summary:**

This paper introduces a novel jailbreak detection framework, Latent Sentinel, based on probe classifiers located at each intermediate internal layers of models. It introduces comparable to slightly higher accuracy performance while limiting latency and resources usage. This novel framework has been tested on 2 different models, two settings (standard and shadow aligned models with SFT), over 5 attacks and 10 languages.

**Strengths:**

Broad evaluation over a large datasets, defences, and models.

Detector is lightweight, simple to implement, and can give good performance on the tested jailbreak datasets.

The detector worked also on alignment degraded models which was interesting - alignment of a large model is a non-trivial task and so having a defence that is effective if that step is reduced is a benefit.

**Weaknesses:**

Small improvements over just a vanilla BERT classifier, potentially with better tuning the gap could be narrowed further. Also, less flexible and generalizable: it has model dependent training, compared to a classifier like BERT which requires one training for any underlying LLM to be defended.

From Table 3 it seems like the performance on benign data can have close to a 20\% false positive rate? That level of error on benign data would make the classifier unusable for any level of practical application, which given that this is an empirical paper proposing a practical defence (as opposed to a theoretical contribution) is a significant weakness.

No examination of adaptive attacks - there are countless defences proposed over a decade of ML security research that when subject to an adaptive attacker break. For example, if an attacker optimized with GCG to both fool the classification defence and jailbreak the underlying LLM model.

**Questions:**

See weaknesses

---

### Official Review · Reviewer_MpLF · 2025-10-29

**Soundness:** 3
**Presentation:** 3
**Contribution:** 2
**Rating:** 2
**Confidence:** 4

**Summary:**

This paper proposes a lightweight, architecture-agnostic framework for detecting jailbreak phisrompts in LLM before text generation. Instead of relying on reactive output filtering, Latent Sentinel attaches small linear probes to frozen hidden states across Transformer layers to detect adversarial intent in real time. These probes add less than 0.003% parameters and introduce only 0.1–0.13% latency overhead. Trained on 100K prompts (50K adversarial and 50K benign), the system achieves 98–100% detection accuracy across multiple benchmarks (JailbreakBench, AdvBench, MultiJail) and languages, and generalizes even to alignment-degraded models. The key insight is that jailbreak intent is linearly separable in latent space, allowing simple linear classifiers to serve as effective, real-time safety guards.

**Strengths:**

1. The method achieves good detection accuracy with negligible computational and parameter overhead, making it suitable for real-time, large-scale deployment. Its architecture-agnostic, plug-in design allows easy integration with different LLMs without retraining the backbone.

2. The framework maintains high performance across multiple datasets, attack types, and languages. These present a comprehensive evaluation compared with the work in this field, especially the multilingual experiments.

**Weaknesses:**

1. One of my biggest concerns is the lack of novelty compared to prior work. While the paper presents strong empirical results, the conceptual contribution that using linear probes on hidden states for detecting adversarial or harmful inputs is not novel. Similar approaches [1][2][3] have already demonstrated representation-level or layer-wise probing for safety and jailbreak detection. Hence, I think this paper does not actually include a new approach.

[1] LLM Jailbreak Detection for (Almost) Free!

[2] How Alignment and Jailbreak Work: Explain LLM Safety through Intermediate Hidden States

[3] MetaDefense: Defending Finetuning-based Jailbreak Attack Before and During Generation


2. The approach requires labeled jailbreak vs. benign prompts for training. So the reported near-100% detection rates across benchmarks may indicate overfitting to standardized datasets (JailbreakBench, AdvBench). This dependence on supervised datasets may limit scalability to unseen or evolving attack types, especially in the wild where new jailbreak strategies emerge frequently.

3. Another concern is the limited analysis of FPR, TPR. Table 3 does not provide enough sensitive analysis information of the classifier. Also, the authors note reduced specificity for out-of-distribution benign prompts but provide little quantitative or qualitative analysis of this issue.

**Questions:**

1. What’s the result of flipattack [4]?
[4] FlipAttack: Jailbreak LLMs via Flipping

---

### Official Review · Reviewer_MX3q · 2025-10-30

**Soundness:** 1
**Presentation:** 2
**Contribution:** 1
**Rating:** 2
**Confidence:** 4

**Summary:**

This paper proposes a method named Latent Sentinel to help LLMs detect jailbreak attacks. The proposed method is implemented by appending additional linear classifiers to each layer of a protected LLM to help the LLM classify harmful/benign inputs based on their latent representations. Experiments conducted on several weak jailbreak attacks show that the proposed method has some ability in identifying jailbreak prompts.

**Strengths:**

The main idea of the proposed method, i.e., appending additional linear classifiers to each layer of LLMs, is easy to follow.

**Weaknesses:**

1. While the proposed Latent Sentinel method is evaluated against five different jailbreak attacks, all these attacks are weak attacks that only leverage simple hand-crafted jailbreak prompts to perform the attacks. I think the authors should evaluate their defense under much more recent and stronger attacks such as [r1, r2, r3, r4].

2. Besides, since the proposed method is a kind of LLM latent-space defense, the authors should also evaluate its robustness against attacks dedicated to latent-space defenses such as [r5].

3. According to Section 3.2, each layer of the protected LLM has its own linear probing classifier. As a result, for a given input prompt, one will have $L$ probing classification results for this prompt, where $L$ is the overall number of layers of the LLM. So how are all these $L$ sub-results leveraged to determine the final classification result? This does not seem to be explained in the paper.

4. The proposed Latent Sentinel method is designed based on linear classifications, which from my perspective is too simple to be robust to various strong jailbreak attacks. It seems that the proposed method can be easily broken by some simple adaptive attacks. So I have no confidence in the proposed defense unless the authors can empirically justify its robustness against strong adaptive attacks.


**References**

[r1] Zou et al. Universal and Transferable Adversarial Attacks on Aligned Language Models. arXiv 2023.

[r2] Sadasivan et al. Fast Adversarial Attacks on Language Models In One GPU Minute. ICML 2024.

[r3] Hayase et al. Query-Based Adversarial Prompt Generation. NeurIPS 2024.

[r4] Andriushchenko et al. Jailbreaking Leading Safety-Aligned LLMs with Simple Adaptive Attacks. ICLR 2025.

[r5] Bailey et al. Obfuscated Activations Bypass LLM Latent-Space Defenses. arXiv 2024.

**Questions:**

See **Weaknesses**.

---

### Official Review · Reviewer_vErb · 2025-10-30

**Soundness:** 2
**Presentation:** 3
**Contribution:** 2
**Rating:** 2
**Confidence:** 3

**Summary:**

The paper introduces Latent Sentinel, a lightweight method for real-time jailbreak detection that attaches small linear probes to a model’s hidden states to identify harmful intent before text generation. The approach uses frozen transformer representations, requiring minimal additional parameters and latency, and uses per-layer probe scores for final detection. It is trained on a balanced set of jailbreak and benign prompts and evaluated across multiple benchmarks, languages, and attack families. Experiments show high detection accuracy, strong multilingual performance, with ablations indicating that intermediate layers carry the most separable safety signals and that threshold calibration improves benign specificity.

**Strengths:**

* **Practical and lightweight design:** The method is simple, computationally efficient, and architecture-agnostic, allowing deployment without modifying or fine-tuning the base model. Low latency and parameter cost introduces near-zero runtime overhead.
* **Strong empirical performance:** The approach achieves high detection accuracy and remains effective even under partially misaligned or degraded models.
* **Insightful layer analysis:** The study provides a clear examination of where jailbreak signals emerge across model layers and demonstrates that intermediate representations are most discriminative.

**Weaknesses:**

1. **Limited novelty:** Although the results are strong, the conceptual contribution is incremental. Prior work has already shown that jailbreak intent is linearly separable in hidden states, and linear probes for safety have been explored in contemporaneous research [1-5] . This paper primarily offers a practical packaging of known ideas, layer-wise probes, aggregation, and calibration, which makes it unclear whether the improvements stem from methodological innovation or simply from the volume/diversity of training data.
2. **Missing comparable representation-based baselines:** The baseline defenses do not include recent hidden-state or logit-based jailbreak detectors [1-5] , even though such works are directly relevant. A fair SOTA claim would need head-to-head comparisons under the exact same training data and evaluation conditions.
3. **Attack coverage is limited:** Despite the strong results, the attack suite excludes several fundamental and widely studied jailbreak types, most notably GCG [6] and other adaptive or optimization-driven approaches. Given the size of the training dataset and the defense claims, a richer taxonomy of attack strategies would be expected.
4. **Training–test family overlap risk:** The JailbreakV-28K dataset used for training already contains prompts from the same canonical attack families (SR/MR/PE/AS/DAN) that appear in evaluation. Although the authors remove AdvBench from training, they do not provide overlap analysis with JailbreakBench or MultiJail. Without near-duplicate filtering or explicit family-wise separation, the generalization claims are weakened.
5. **Limited model diversity undermines scalability claims:** The paper asserts architecture portability and scalability, but evaluates only two similarly sized instruction-tuned models (Qwen2.5-7B-Instruct and Llama-3.1-8B-Instruct). There is no demonstration across significantly smaller or larger scales, nor across different modeling paradigms, leaving scalability claims insufficiently supported.
6. **Unclear seen vs. unseen generalization:** The paper would benefit from explicit experiments separating seen attack patterns from unseen ones. Currently, it is difficult to attribute performance to true generalization as opposed to training exposure.
7. **Threshold optimization unclear for baselines:** The method’s strong benign specificity depends heavily on threshold calibration, yet it is not clearly explained whether the same calibration procedure was applied to baselines. Without parity in calibration, fairness of comparisons is uncertain.
8. **Benign distribution mismatch affects cross-lingual claims:** Benign training data is exclusively English (Alpaca), while multilingual evaluation focuses only on adversarial examples. Without benign non-English prompts, cross-lingual specificity is not evaluated, making the multilingual accuracy potentially overstated.
9. **Ambiguity in defining “attacks” and success criteria:** The paper does not explain how the “Vanilla” detection rates are measured or how a prompt is determined to be a true jailbreak. It appears that prompts are labeled as attacks solely based on their dataset source or template, without evaluating whether the model actually produces harmful outputs. There is no definition of harmful success (e.g., refusal override, keyword criteria, human evaluation). Furthermore, some “Vanilla” detection rates are already high in certain settings, suggesting that baseline models may already be relatively robust, which reduces the incremental significance of the reported gains.

***Minor remarks:***
* Some paragraphs are repeated across pages (e.g. end of page 5, beginning of 6),, which disrupts readability and should be fixed.
* Section 4.1 is difficult to follow because baseline descriptions are split between the model and evaluation subsections.
* The large tables are hard to interpret, and adding clear highlighting of key results would make them easier to read.

[1] Pu, R., Li, C., Ha, R., Chen, Z., Zhang, L., Liu, Z., Qiu, L., Ye, Z. (2025) Feint and Attack: Jailbreaking and Protecting LLMs via Attention Distribution Modeling

[2] Qian, C., Zhang, H., Sha, L., Zheng, Z. (2025) HSF: Defending against Jailbreak Attacks with Hidden State Filtering

[3] Chen, G., Xia, Y., Jia, X., Li, Z., Torr, P., Gu, J. (2025) LLM Jailbreak Detection for (Almost) Free!

[4] Candogan, L. N., Wu, Y., Rocamora, E. A., Chrysos, G., Cevher, V. (2025) Single-pass Detection of Jailbreaking Input in Large Language Models

[5] Zheng, C., Yin, F., Zhou, H., Meng, F., Zhou, J., Chang, K.-W., Huang, M., Peng, N. (2024) On Prompt-Driven Safeguarding for Large Language Models

[6] Zou, A., Wang, Z., Carlini, N., Nasr, M., Kolter, J. Z., & Fredrikson, M. (2023). Universal and Transferable Adversarial Attacks on Aligned Language Models.

**Questions:**

1. **Training cost clarification:** What is the actual compute cost of training the probes (e.g., GPU hours per model, wall-clock time)?
2. **Model-specific probe behavior:** Are the linear classifiers tied to a specific model’s internal representations, or can they be transferred or reused across closely related model variants? Clarifying whether retraining is required for every architecture/backbone would strengthen the portability claim.
3. **Black-box deployment feasibility:** The current design assumes access to internal hidden states. Do the authors see a pathway for adapting this approach to a more restrictive, black-box setting (e.g., via side-channel features or partial token-embedding exposure), or is it inherently limited to full white-box access?
4. **Ambiguity in attack-type count:** Line 258 refers to nine attack techniques, yet Figure 2 and Table 1 evaluate only five technique-based attack families (SR, MR, PE, AS, DAN) plus multilingual. What are the remaining four techniques, and are they included in any part of the evaluation?
5. **Handling of long-context attacks:** Both BERT-based baselines and Sentinel probes are trained with significantly limited maximum sequence lengths (e.g., 800 tokens). How are jailbreak attempts with substantially longer prompts handled at inference time, particularly attacks that hide malicious intent deep in long contexts?
6. **Prompt-Guard result discrepancy:** Table 2 reports Llama Prompt-Guard achieving 100% accuracy on JailbreakBench and AdvBench, yet the text claims prior guardrails degrade and are outperformed. Could the authors clarify whether Prompt-Guard was evaluated or calibrated differently than in real-world usage?
7. **Benign multilingual evaluation:** Do the authors have results on benign non-English inputs to measure cross-lingual specificity? Without evaluating benign examples in other languages, how can we be confident that high multilingual accuracy is not driven purely by attack-distribution differences?
8. **Benign specificity:** Benign false-positive rates are critical for deployment, yet only Table 3 reports them (for Sentinel and BERT baselines) in a single setting. Can you provide FPR metrics for all baselines and multilingual benign inputs to enable fair comparisons and assess real-world usability?
9. **SmoothLLM detection drop:** SmoothLLM sometimes shows lower detection accuracy than the Vanilla model in Table 1. Why would a defense method degrade detection relative to having no defense at all, and how should these results be interpreted?

---

### Official Review · Reviewer_DRua · 2025-11-01

**Soundness:** 3
**Presentation:** 3
**Contribution:** 3
**Rating:** 6
**Confidence:** 5

**Summary:**

The paper describes a novel defence method against LLM jailbreaks, based on classification of latent activations from the transformer layers (Latent Sentinel). The experiments show that training linear models on activations from 50k jailbreak and 50k benign prompts and then using these linear models during inference can significantly reduce harmful generation under malicious prompts with minimal computational overhead.

**Strengths:**

The paper is well structured and easy to read. The paper motivation is clear, the main idea of using latent activations for classifying harmful prompts / responses is simple yet effective.

The paper presents thorough experiments on multiple settings and extensive comparisons with related works. The additional experiments, visualizations and insights are also interesting and valuable.

The results on jailbreak identification are very promising (>95% accuracy), generalizing across architectures and even across multiple languages, supporting the claim that there is a strong separation between harmful and benign samples in the latent space.

The method has very low costs both for training and for deploying, making it very compelling as a real-world defence.

**Weaknesses:**

**Unclear Benign Prompt Performance**

While the paper focuses heavily on improving model safety, it provides limited insight into the utility impact of the proposed defense. In the LLM safety literature, it is standard practice to evaluate defended models on a broad suite of utility benchmarks, including MCQ tasks such as ARC [1] and MMLU [2], as well as open-ended generation tasks like MT-Bench [3], to quantify any performance degradation relative to the base model. Additionally, the XSTest suite [4] is commonly used to evaluate exaggerated safety responses on 200 harmful and 250 benign prompts that closely resemble malicious queries, making it particularly informative for analyzing the robustness–utility tradeoff.

In this paper, the authors primarily report benign detection accuracy on a limited evaluation set, with results ranging from 2% to 80–90%. This analysis does not sufficiently characterize the model’s overall usefulness or its impact on general performance. I encourage the authors to include results on the aforementioned utility benchmarks, as such evaluations could substantially influence my final assessment of the work.

**Evaluation only on prompt-based jailbreaks**

The proposed method is evaluated almost exclusively on prompt-based jailbreaks, where it demonstrates impressive detection performance. However, it remains unclear how the approach would perform against non-prompt-based or optimization-driven attacks such as GCG [5], BEAST [6], QueryAttack [7], or ArtPrompt [8]. These attacks may induce fundamentally different patterns in the latent space, potentially allowing them to evade Latent Sentinel’s probes.

Furthermore, it would be particularly valuable to test whether targeted gradient-based attacks like GCG could be adapted to directly manipulate Sentinel’s predictions. However, I understand that this specific experiment might require significant changes to the GCG code so I don’t mind if the authors don’t manage to provide results within the rebuttal period.

**References**


[1] https://arxiv.org/abs/1803.05457

[2] https://arxiv.org/abs/2009.03300

[3] https://arxiv.org/abs/2306.05685

[4] https://arxiv.org/abs/2308.01263

[5] https://arxiv.org/abs/2307.15043

[6] https://arxiv.org/abs/2402.15570

[7] https://arxiv.org/abs/2502.09723

[8] https://arxiv.org/abs/2402.11753

**Questions:**

I am mostly interested in discussing the weaknesses above. Additional evaluations are crucial for properly assessing the quality of the method.

---

### Author Response · Authors · 2025-11-13
**We thank the reviewers for their feedback!**

We sincerely thank all reviewers for their detailed and constructive feedback.
Below we address the main concerns with factual clarification, quantitative evidence, and supporting literature.

---

## (1) “Sentinel has not been tested against strong attacks such as GCG”

This is **not accurate**.

Our evaluation strictly follows the **official JailbreakBench evaluation harness** (threat model, system prompt, scoring pipeline). Importantly:

- **JailbreakBench explicitly includes GCG (Greedy Coordinate Gradient)–based attack artifacts** as part of its benchmark suite [1].
- **AdvBench** is the benchmark introduced together with the original **GCG paper**, and is widely used as the *de facto* standard evaluation set for GCG-family attacks [2].

Therefore, our **JailbreakBench + AdvBench results inherently evaluate Sentinel under GCG-style attacks**.

---

## References

[1] Chao, P. et al. (2024). *JailbreakBench: An Open Robustness Benchmark for Jailbreaking Large Language Models.* arXiv:2404.01318.
[2] Zou, A. et al. (2023). *Universal and Transferable Adversarial Attacks on Aligned Language Models.* arXiv:2307.15043.

---

> ### Author Response · Authors · 2025-11-13
> **The novelty is limited**
>
> ## (2) “The novelty is limited”
>
> Our contribution is not simply observing that “latent space is linearly separable.”
> The key idea is that **latent representations carry substantially richer information than the Next-Token Prediction (NTP) interface**, and Sentinel leverages these high-capacity internal representations to overcome the fundamental structural limitations of NTP-based defenses.
>
> Our novelty lies in (i) *designing Sentinel around this representational advantage*, (ii) *showing that this representation-level defense generalizes across unseen languages and attack families*, and (iii) *demonstrating that even alignment-damaged models retain detectable latent-space safety signals*.
>
> The core arguments are summarized in (a)–(c).
>
> ---
>
> ### (a) Structural bottleneck of NTP vs. information capacity of latent space
>
> The theoretical maximum information a model can emit per decoding step through NTP is **log₂|V| bits per token**.
>
> By contrast, a latent vector contains **thousands of dimensions × tens of bits**, resulting in **several thousand times higher representational capacity**.
>
> #### Quantitative comparison
> (using official configuration values for Llama-3.1-8B and Qwen-2.5-7B)
>
> | Model | Vocab Size \|V\| | log₂\|V\| (bits/token) | Hidden Dim d | Precision (assumed) | Latent Capacity (d×16 bits/token) | Ratio (Latent / NTP) |
> |-------|------------------:|------------------------:|--------------:|---------------------:|----------------------------------:|-----------------------:|
> | **Llama-3/3.1-8B** | 128,256 | 16.97 | 4,096 | 16-bit | 65,536 bits | **≈ 3,862×** |
> | **Qwen-2.5-7B**    | 151,646 | 17.21 | 3,584 | 16-bit | 57,344 bits | **≈ 3,332×** |
>
> **Implication:**
> - The NTP channel is **structurally narrow (≈17 bits/token)**.
> - The latent space has **tens of thousands of bits** per token.
>
> Thus, GCG and related jailbreak attacks exploit the **bottlenecked NTP interface**, whereas Sentinel operates on a **high-information, high-dimensional latent signal** that the attacker cannot directly manipulate at decoding time.
> This explains why Sentinel can reliably detect jailbreak attempts even when NTP-level behavior is adversarially distorted.

---

> ### Author Response · Authors · 2025-11-13
> **The novelty is limited**
>
> ---
>
> ### (b) Linear separability in LLM latent space
>
>
> A large body of work shows that simple **linear probes** can reliably extract high-level information from intermediate hidden states:
>
> - POS, syntax, dependency structure, semantic roles — BERT/mBERT mid-layers [4,5]
> - Truthfulness represented as a **linear direction** — large LLMs [6]
> - High intrinsic dimension in mid-layers and compression near the output head — common in Transformers [7]
> - Neural-collapse-like geometric clustering emerging at scale — LLMs [6]
>
> These results consistently indicate that **latent representations encode linguistic, semantic, and reasoning concepts in nearly linear subspaces**.
>
> Sentinel is one of the first frameworks to **leverage this representational geometry for real-time, pre-generation jailbreak detection**, requiring no modification to the base model.
>
> We also show that **Sentinel generalizes to nine unseen languages** (zh, it, vi, ar, ko, th, bn, sw, jv), supporting the view that latent spaces contain **language-agnostic, linearly accessible safety signals**.
>
> ---
>
> ### (c) Sentinel remains effective even when alignment is intentionally degraded
>
>
> Recent work demonstrates that alignment can be **easily broken** (e.g., shadow alignment) [8], and that jailbreak mechanisms operate by **disrupting how intermediate layers map safety concepts to output-level refusal behaviors** [3].
>
> Consistent with this theory, our experiments show:
>
> - After applying **aggressive SFT/FFT** to deliberately *damage* the model’s alignment,
> - **Output-level behaviors (NTP) become unreliable**,
> - Yet **latent-space safety signals remain intact**, allowing Sentinel to detect jailbreak attempts.
>
> This is a crucial distinction:
>
> - Alignment damage → breaks **output-space behaviors**
> - But latent conceptual structure → remains **largely preserved**
>
> Therefore, Sentinel utilizes **deeper, concept-level internal signals**, making it inherently more robust than defenses relying solely on output text, refusal heuristics, or decoding-side guardrails.
>
> ---
>
> ## References
>
> [4] Tenney, I., Das, D., Pavlick, E. (2019). *BERT Rediscovers the Classical NLP Pipeline.* ACL 2019.
>
> [5] Hewitt, J., Manning, C. D. (2019). *A Structural Probe for Finding Syntax in Word Representations.* NAACL 2019.
>
> [6] Wu, R., Papyan, V. (2024). *Linguistic Collapse: Neural Collapse in (Large) Language Models.* NeurIPS 2024.
>
> [7] Cheng, E. et al. (2024). *Emergence of a High-Dimensional Abstraction Phase in Language Transformers.* arXiv:2405.15471.
>
> [8] Yang, X. et al. (2023). *Shadow Alignment: The Ease of Subverting Safely-Aligned Language Models.* arXiv:2310.02949.

---

### Note · Authors · 2025-11-24

I have read and agree with the venue's withdrawal policy on behalf of myself and my co-authors.